# The Indigenous Adolescent Oral Health Partnership Study: A Co-Design Study Protocol

**DOI:** 10.3390/ijerph19159104

**Published:** 2022-07-26

**Authors:** Zac Calvin, John Skinner, Yvonne Dimitropoulos, Gabriela Stan, Julie Satur, Susan Cartwright, Richard P. Widmer, Tiarnee Schafer, Rachel Williams, Woosung Sohn, Sarah Raphael, Bradley Christian, Carmen Parter, Lauren Blatchford, Boe Rambaldini, Stephanie R. Partridge, Elyse Cain, Kylie Gwynne

**Affiliations:** 1Faculty of Medicine and Health, The University of Sydney School of Medicine, Anderson Stuart Building, The University of Sydney, Sydney, NSW 2006, Australia; zcal2329@uni.sydney.edu.au; 2Poche Centre for Indigenous Health, Room 224 Edward Ford Building, The University of Sydney, Sydney, NSW 2006, Australia; john.skinner@mq.edu.au; 3Centre for Global Indigenous Futures, Macquarie University, 3/75 Talavera Road, Macquarie Park, NSW 2113, Australia; boe.rambaldini@mq.edu.au (B.R.); kylie.gwynne@mq.edu.au (K.G.); 4Department of Linguistics, Macquarie University, 16 University Avenue, Macquarie Park, NSW 2113, Australia; 5Kingswood TAFE, 12–44 O’Connell Street, Kingswood, NSW 2747, Australia; gabriela.stan2@tafe.nsw.edu.au; 6Melbourne Dental School, Level 5, 720 Swanston Street, The University of Melbourne, Melbourne, VIC 3010, Australia; juliegs@unimelb.edu.au; 7Colgate-Palmolive Company, Level 14, 345 George Street, Sydney, NSW 2001, Australia; susan_cartwright@colpal.com; 8Paediatric Dentistry, The Children’s Hospital at Westmead, Corner of Hawkesbury Road and Hainsworth Street, Westmead, NSW 2145, Australia; richard.widmer@health.nsw.gov.au; 9Black Dog Institute, Hospital Road, Randwick, NSW 2031, Australia; tsch5495@uni.sydney.edu.au; 10Armajun Aboriginal Health Service, 1 Rivers Street, Inverell, NSW 2360, Australia; rwilliams@armajun.org.au; 11Sydney Dental School, Faculty of Medicine and Health, The University of Sydney, Darcy Road, Westmead, NSW 2145, Australia; woosung.sohn@sydney.edu.au; 12Australian Dental Association NSW Branch, L1 1 Atchison Street, St Leonards, NSW 2065, Australia; sarah.raphael@adansw.com.au; 13Western NSW Local Health District, Poplars Building, Bloomfield Campus, Forest Road, Orange, NSW 2800, Australia; bradley.christian@health.nsw.gov.au; 14Poche Centre for Indigenous Health, 31 Upland Road, University of Queensland, St Lucia, QLD 4067, Australia; c.parter@uq.edu.au; 15Albury Wodonga Aboriginal Health Service, 664 Daniel Street, Glenroy, NSW 2640, Australia; lauren@awahs.com.au; 16Engagement and Co-Design Hub, Faculty of Medicine and Health, The University of Sydney, Level 6, Block K, The University of Sydney at Westmead Hospital, Westmead, NSW 2145, Australia; stephanie.partridge@sydney.edu.au; 17NSW Council of Social Service, Level 3, 52–58 William Street, Woolloomooloo, NSW 2011, Australia; elyse@ncoss.org.au

**Keywords:** co-design, oral health, adolescent, Australian, Aboriginal and Torres Strait Islander

## Abstract

Background: in this protocol we outline a method of working alongside Aboriginal communities to learn about and facilitate improvement in the oral health habits in Aboriginal adolescents. By facilitating positive oral health in Aboriginal adolescents, we hope to achieve lifelong improvement in oral health and general wellbeing. Methods: this paper outlines a co-design methodology through which researchers and Aboriginal communities will work together to create a custom oral healthcare program aimed at Aboriginal adolescents. Researchers, a youth advisory group, Aboriginal community-controlled health services and three regional NSW communities will together devise an oral health strategy focused on five components: application of topical fluoride, increasing water consumption, improving nutrition, daily toothbrushing, and enhancing social and emotional wellbeing. Capacity building is a key outcome of this program. Discussion: as the gap in health status between Aboriginal and non-Aboriginal people remains wide, it is clear that new approaches and attitudes are needed in Aboriginal public health research. This protocol is representative of this shifting approach; giving power to Aboriginal communities who seek to have sovereignty and self-determination over their healthcare. Trial registration: TRN: ISRCTN15496753 Date of registration: 20 October 2021.

## 1. Introduction

In Australia, a gap exists in health outcomes between Aboriginal and Torres Strait Islander people (hereafter referred to as Aboriginal people) and other Australians. Many studies examine and quantify the facets of ill-health in Aboriginal people [1]. This deficit discourse in the literature, without offering culturally safe solutions, contributes to feelings of shame and stigma among Aboriginal communities [1]. New approaches are needed which use the strengths of Aboriginal communities to collaboratively produce health promotion programs to positively impact the determinants of health for Aboriginal people.

Health-related behaviours can be major causes or preventers of serious morbidity. Health-related behaviours that begin in adolescence affect a person’s health both in the current stage of life and throughout adulthood [2]. This protocol defines adolescence as ages 10–19 in line with the World Health Organisation definition [3]. Oral health and oral hygiene behaviours also contribute to overall wellbeing and have a substantial influence on overall health [4]. Periodontal (gum) disease is associated with many systemic conditions including cardiovascular disease [5], pneumonia and kidney disease [6]. A strong bidirectional relationship also exists between periodontal disease and diabetes [7].

In terms of oral health, Aboriginal adolescents have a greater average number of dental caries and receive more frequent tooth extractions compared to their non-Aboriginal peers [8,9]. Aboriginal people are less likely to receive timely preventative dental treatment [10] and are more likely to be hospitalised due to oral health infections and disease [8]. This can be attributed to many of the social and cultural determinants of health [10]. Few studies have effectively examined or sought to change oral health outcomes for Aboriginal adolescents during these formative years [11].

A recent systematic review found that successful oral health interventions for Indigenous people globally adopted culturally safe, participatory and collaborative approaches [12]. It concluded that an effective program should employ local workers, utilise multiple settings and strategies and address the social, economic and cultural determinants of health [12]. Despite this, few oral health research papers have explored social determinants of health in this context or established career development pathways for local workers despite many claiming to adopt a ‘capacity building’ approach [12].

High strength fluoride varnishes given to children two to four times per year are associated with a significant reduction in the development of future dental caries by up to 43% [13]. In New South Wales (NSW), high strength fluoride varnish may be applied by dentists, dental therapists, oral health therapists and dental hygienists [14]. Non-registered oral health workers such as dental assistants may apply fluoride varnish following specific training and approval from relevant regulatory authorities [15]. A previous study demonstrated the feasibility of utilising oral health therapists to apply fluoride regularly to Aboriginal children in rural schools in NSW [16]. This study also noted the employment of dental assistants to apply fluoride varnish. This would improve sustainability, build community capacity, and increase potential reach of such a program which provides an important link to this study [16]. The application of fluoride varnish together with locally designed strategies to encourage tooth brushing will be critical when designing preventative health promotion initiatives.

Tooth brushing is a central component of achieving and maintaining good oral health. Despite this, only 54.5% of adolescents in NSW report that they brush their teeth twice or more per day [17]. Furthermore, adolescents in NSW who brush their teeth twice a day have fewer decayed, missing or filled teeth compared to those who do not (0.91 vs. 1.46 teeth) [17]. A survey in Central Northern NSW found that only 12.8% of Aboriginal children aged 5–12 years reported brushing their teeth the morning before the survey and 64.1% reported owning a toothbrush at home [18]. A co-designed, school-based program was effective at increasing rates of tooth brushing among children in the region [19]. The authors noted that community ownership was important in ensuring the success and sustainability of the program [19]. Despite this success, no subsequent studies have sought to adapt a tooth brushing initiative aimed towards Aboriginal adolescents in Australia. Continuity of tooth brushing, and dental health education is needed to ensure Aboriginal adolescents maintain good oral health habits into adulthood.

The consumption of sugar-sweetened beverages is associated with an increased risk of dental caries in children and poorer metabolic health, and high among Aboriginal adolescents [20]. Adolescents in NSW who regularly consumed one or more sugary drinks per day had a greater number of decayed missing or filled teeth compared to those who did not (1.21 vs. 0.8 teeth) [17]. The Australian dietary guidelines recommend the consumption of fluoridated tap water and the limiting of sugar-sweetened beverages such as soft drinks, cordials, fruit juices and sports drinks [21]. Australian rural communities experience high temperatures in summer months. Taste aversion, poor water quality and inaccessibility to free refrigerated and filtered water can leads adolescents to opt for store-bought sugar sweetened beverages over tap water [19]. A study in Central Northern NSW installed refrigerated and filtered water fountains in key community locations as part of a suite of preventive strategies to improve oral health [19]. As a result, children reported consuming fewer sugar-sweetened beverages [19]. Novel approaches such as this are needed to increase the consumption of fluoridated tap water and decrease the consumption of sugar-sweetened beverages.

Dietary risk factors are estimated to contribute to 9.7% of the burden of disease in Australia’s Aboriginal population [22]. Among Aboriginal Australians, high school-aged adolescents were the least likely to meet the National Health and Medical Research Council (NHMRC) guidelines for consumption of fruit and vegetables [23]. In 2012–2013 only 21% of Aboriginal adolescents aged 15–17 years met the dietary guidelines for adequate fruit intake while 2.8% met the recommended intake of vegetables [23]. This indicates that Aboriginal adolescents may require additional support to meet dietary guidelines. Diet and eating patterns play a significant role in the development of oral disease thus dietary education should be included in a successful oral health program [24]. A 2013 Japanese study found participation in a dietary education program in high school correlated with a reduction in dental caries in early adulthood [25].

The current standard of health service delivery has been ineffective in closing the gap between the health outcomes of Aboriginal and non-Aboriginal people [26]. Health programs designed for the general population often fail to meet the needs of Aboriginal communities, especially adolescents. Effective healthcare for Aboriginal people must have a consistent focus on cultural competence and strong organizational, community and clinical governance [26]. The National Aboriginal and Torres Strait Islander Health Implementation plan also outlines the strategic need to involve adolescents in the planning and implementation of strategies addressing their health needs [27]. Co-design is a process which requires active involvement of all stakeholders. Co-design is not merely consultation or participant approval; but a methodology where the beneficiaries of the project or research participate in every step of the process from conception to design, implementation and evaluation [28]. In the context of Aboriginal health, co-design seeks to ensure a health program is created which is responsive to the unique needs of local Aboriginal people [28]. Co-design also helps gain necessary collective buy-in of the community and leads to higher rates of uptake and involvement and thus contributes to improved health outcomes [28]. This paper describes the protocol for a co-designed, culturally safe, evidence-informed approach to improving the oral health of Aboriginal adolescents within a wider social determinants approach.

## 2. Materials and Methods

The Indigenous Adolescent Oral Health Partnership Study is an implementation science project using a co-designed, multidisciplinary, mixed methods approach in collaboration with Aboriginal communities in NSW, Queensland and Victoria, Australia. It involves a partnership between tertiary education institutions, Aboriginal Community Controlled Health Services (ACCHS), professional organisations, industry and health care providers. Partners will provide expertise, in-kind support or guidance throughout the study. The study will comprise three broad phases including: planning, implementation, and analysis. Co-design methodology will be embedded throughout each phase.

In 2019, a video was produced by the Poche Centre for Indigenous Health at the University of Sydney with Aboriginal people from NSW who were undertaking vocational training in Dental Assisting. The purpose of this video was to promote participation in a questionnaire to identify possible oral health promotion strategies for Aboriginal adolescents. Prior to participating in the questionnaire, participants were issued a study information statement and signed a consent form. The results of this pre-pilot questionnaire, as well as recommendations of a literature review examining successful components of oral health interventions for Indigenous adolescents globally, have led to the development of five core oral health aims for this study. These include: (1) increasing topical fluoride application; (2) increasing tooth brushing (3) increasing the consumption of water; (4) improving nutrition behaviours; and (5) enhancing social and emotional well-being among Aboriginal adolescents in communities in NSW, Queensland and Victoria. This study will take place over five years (2022–2027) and across three phases. These phases include (1) Planning (2022–2023); (2) Implementation (2023–2025); and (3) (2025–2027). Figure 1 demonstrates the three phases of this study.

We hypothesise that involving Aboriginal and Torres Strait Islander adolescents in the design and delivery of evidence-based oral health promotion strategies will improve the oral health and social and emotional wellbeing of Aboriginal and Torres Strait Islander adolescents.

### 2.1. Planning Phase

The planning phase will involve the establishment of a youth advisory group (YAG) to guide the study at each stage, meeting quarterly over the two-year study period. The YAG will be established by the Poche Centre for Indigenous Health in conjunction with representatives from The Matilda Centre for Research in Mental Health and Substance Use, the NSW Aboriginal Health and Medical Research Council (AH&MRC) and the Black Dog Institute; as well as eight Aboriginal people aged 10–19 years from Aboriginal communities across Australia.

During this phase, Aboriginal people undertaking vocational training in Dental Assisting (Qualification: Certificate IV in Dental Assisting [oral health promotion]) will be invited to be co-researchers in this study. As co-researchers they will be offered to complete a separate vocational qualification on research in communities (Qualification: Skill Set in Community Research). This qualification will cover basic research methodologies and data collection to enable co-researchers to collect data systematically.They will also be trained in the application of fluoride varnish and assist in the development of oral health promotion strategies such as tooth brushing and toothpaste distribution that will be included in the implementation phase of the study, under the supervision and guidance of oral health practitioners from the local ACCHS who have partnered with this study.

Co-researchers will conduct co-design workshops as part of the planning phase within the three participating communities in NSW, Queensland and Victoria to enable Aboriginal adolescents, families, community members and Elders to work with researchers to develop and refine an oral health program that is specific to the needs and customs of their particular community. The co-design process will likely include yarning and other Indigenous research methodologies. The oral health program will include key strategies that are based on the five core oral health aims of this study.

### 2.2. Implementation Phase

During the implementation phase, the co-researchers will implement all five strategies within their participating community. The co-design workshops will determine how each strategy is implemented in each community as this will be specific to the needs of each community.

The first strategy will involve the quarterly application of fluoride varnish to Aboriginal adolescents by Aboriginal people who are completing their training in Dental Assisting. The application of fluoride varnish will be under the guidance of an oral health practitioner from the local ACCHS or the Poche Centre for Indigenous Health. The setting of this strategy will be subject to co-design workshops, meeting the needs and wants of the community.

The second strategy will target toothbrushing with a fluoride toothpaste. Participating Aboriginal adolescents will be provided with high quality electric toothbrushes and fluoride toothpaste. High strength fluoride toothpastes may be considered at an individual level and, if provided, safe storage and application instructions will be provided. A combination of toothbrushing apps, SMS reminders, competition leader boards and prizes will be used to encourage regular twice daily brushing. The composition of effective and encouraging reminders and prizes will be subject to co-design workshops held in each community.

The third strategy will target water consumption among Aboriginal adolescents. The co-design workshops will discuss the installation of filtered and refrigerated water fountains with usage monitors in each community. The co-design workshops will enable the local Aboriginal community, including participating Aboriginal adolescents to identify suitable locations for these fountains and may include local schools, sports grounds, or youth recreation areas. Water fountains installed as part of this program, although filtered, will not remove any fluoride from the water. High quality, durable aluminium water bottles will be provided to students at the local secondary school. Participating Aboriginal adolescents will be invited to design artwork to feature on the water bottles and fountains.

The fourth strategy will involve Aboriginal people who are completing their studies in Dental Assisting facilitating group education sessions with participating Aboriginal adolescents to discuss the causes and prevention of oral disease, how dietary habits impact oral health and how to enact behavioural change to improve diet and oral health. The co-design workshops will determine where these sessions will take place and their frequency.

The fifth strategy will aim to enhance the social and emotional wellbeing of participating Aboriginal adolescents and involve a peer-led buddy program. An Aboriginal person completing their studies in Dental Assisting will pair with a participating Aboriginal adolescent to provide advice and support. We will encourage the continuation of this peer led buddy program after the conclusion of the study. Culturally safe professional psychological support will be available, should they be needed at any time through the duration of the study. The specific methods of providing social and emotional support will be determined at the co-design workshops, to ensure any support provided is culturally competent.

Utilising co-design methodology throughout the implementation phase will ensure that each strategy is implemented in such a way that remains culturally competent, sustainable and meets the needs and wants of a specific community. This process allows for the community and investigators to make changes to the oral health program as additional strengths or weaknesses of each strategy become apparent.

### 2.3. Setting

This study will take place in Aboriginal communities in NSW, Queensland and Victoria. The precise setting for each strategy will be subject to the YAG and co-design workshops to maximise reach, convenience and suitability for participants. Settings may include the health service, community centres, online forums, local schools or sports clubs.

### 2.4. Co-Researchers

Capacity building is a core aspect of this study. Local Aboriginal oral health workers from the participating ACCHS and members of the YAG will be engaged as co-researchers in the oral health program. Furthermore, Aboriginal people completing their studies in Dental Assisting at TAFE NSW Western Sydney Institute will also be co-researchers, assisting in the development and implementation of the strategies and hosting the co-design workshops within each community.

Specifically, the courses and vocational training units of competency Aboriginal people will be completing will include: Certificate IV in Dental Assisting and ‘Implement an oral health promotion program’ (HLTDEN011); ‘Implement an individualised oral hygiene program’ (HLTDEN004) and ‘Apply fluoride varnish’ (HLTOHC006) and a Skill Set in Community Research. Participation in this study and developing and implementing the oral health promotion program will ensure students have experience and competency upon graduation.

### 2.5. Recruitment of Participants

Aboriginal adolescents aged 10 to 19 years will be eligible to participate in this study. Participants must reside within the catchment of the partnering ACCHS and are expected to continue living there for the duration of the study. A target number of 60 participants has been set for this study. This number was agreed on to enable a diverse range of participants, including age, gender and location. Partnering ACCHSs will be active in the recruitment of participants. Information promoting the study will be placed in waiting rooms of the health service and posted on the social media page of the ACCHS to engage participants in a culturally appropriate way. When an Aboriginal adolescent presents to a medical or dental consultation, they and their parent or guardian will be informed of the study and invited to participate by the local co-researcher. The co-researcher will provide written and verbal participant information and seek written consent from their parent or guardian for their participation in the study. Participants will be advised that participation in the study is entirely voluntary and will not affect their relationship with the ACCHS should they wish to cease their involvement in the study.

### 2.6. Data Collection

Baseline oral health status of participants will be recorded by the co-researcher from the local ACCHS. A co-researcher who is a registered dental practitioner will record the number of teeth and the presence of dental caries into the participant’s electronic medical record using Titanium dental software (Titanium Solutions, Auckland, New Zealand). Participating adolescents will also be asked to complete a questionnaire to determine their baseline oral health habits. This questionnaire will be administered by the Aboriginal co-researchers and collect information on the participant’s demographics, perceived overall oral health, oral hygiene routine, brushing frequency, consumption of sugar-sweetened beverages and past dental treatments. In order to minimise recall bias an existing validated oral health questionnaire will be used as a guide, We will work with the YAG to ensure the questionnaire is culturally safe. The dental screening and questionnaire will also be completed at the conclusion of the oral health program for comparison. Data collected will also be compared to any locally available individual clinical data with participant or guardian consent. Comparison of summary data will also be made with existing state and national data sets on oral health and hygiene behaviours of adolescents.

Co-researchers will also collect data on the uptake and efficacy of each strategy. This will include the number of fluoride varnish applications for each participant, utilisation of the toothbrush tracking program and water fountain usage meter readings. The co-design methodology will allow co-researchers, participating dental assisting students and participating adolescents to provide input on strengths and weaknesses of the oral health program continuously and make changes to the protocol as required to ensure the implementation of strategies remains relevant and meets the needs of the local community.

### 2.7. Data Holding Policy

Data collected throughout this study will remain the intellectual property of the local participating ACCHS. Health data of participants will be stored securely in Titanium electronic medical records of the respective ACCHS. The publishing of this research may only occur after consent has been given by the ACCHS and the collaborating Aboriginal communities.

### 2.8. Analysis Phase

Quantitative data on oral health status and oral health habits of participating adolescents will be analysed using statistical analysis software (IBM SPSS Statistics Version 26) to examine and quantify changes in oral health status at the completion of the oral health program. These results will also be compared with NSW and National figures from the NSW Teen Dental Survey and the National Child Oral Health Survey. Qualitative data regarding utilisation and engagement will be analysed thematically to determine the overall utility and acceptance of each section of the oral health program. This analysis will inform the future expansion, development or removal of components of the oral health program and recommendations for future oral health promotion strategies targeting Aboriginal adolescents.

### 2.9. Dissemination of Results

The results of this study will be reported back to the participating communities. The research team will use social media, written bulletins, posters and community presentations to disseminate the study findings and implications in culturally safe language. Public events will allow community members to ask questions of the research team and gain an understanding of the health status and health improvements made in their community through the study. This may provide benefit of the oral health promotion programs to members of the community who were not direct participants in the study.

The study results will be published in a reputable peer-reviewed scientific journal and presented at research symposiums. Members of the research team including lead investigators, ACCHS staff, co-researchers and dental assistant students will contribute to these journal articles and presentations. The full breadth of study contributors, participants and community facilitators will be acknowledged for their work in co-designing and implementing this project. The valuable contribution of Aboriginal cultural knowledge will be highlighted in these presentations.

## 3. Discussion

This protocol describes a co-designed implementation project developed in close partnership with several NSW ACCHSs with a key focus on enhancing social and emotional wellbeing. The NHMRC sets out six core values to conduct ethical research with Aboriginal and Torres Strait Islander Peoples and communities including: spirit and integrity, cultural continuity, equity, reciprocity, respect and responsibility [26]. The methods used in this project and the grounding of a co-design approach ensures the study adheres to these six core values and ensures the research project will be culturally safe for Aboriginal people. The project will ensure cultural continuity with the strong involvement of Aboriginal viewpoints and ideas from start to finish via the pre-pilot questionnaire, YAG, co-design workshops, implementation and reporting back results. The embedded co-design methodology throughout this study will allow us to recognise the importance of collective decision making in Aboriginal communities whilst also demonstrating understanding of the diversity of perspectives between Australia’s many different Aboriginal peoples. The recruitment of Aboriginal dental practitioners and Aboriginal dental assistant students as co-researchers brings a greater array of relevant cultural knowledge to the research and implementation team, as well as capacity-building.

Improving the health outcomes and inequity in healthcare access of Aboriginal adolescents is central to this study. This study aims to create a balance of power between the research team and the Aboriginal community through our methods of co-design, data-holding policy and requirement of final community consent to publish. Participants will be advised of their rights and powers under the project’s governance principles during a robust, informed consent process. Furthermore, all investigators will undergo cultural-safety training to encourage awareness of their own beliefs and cultural differences to ensure a respectful research environment. An equal balance of power between investigators and participants helps researchers to act responsibly and in the best interest of Aboriginal adolescents and their community. As the study is centered on the vulnerable population of Aboriginal adolescents, robust risk management strategies will be in place to ensure researchers are able to be responsive to the needs of participants and prevent harm, should issues arise.

This study will provide an immediate benefit to participants and the wider community through preventative healthcare (fluoride varnish, health education), as well as health-promoting environment (toothbrushes, water fountains) and capacity building including scholarships for local Aboriginal people to complete vocational training in Dental Assisting and employment of local oral health workers as co-researchers. Members of the YAG will be reimbursed (using a recognized reimbursement policy) for their time and input to the project. The project will enable the students completing their studies in Dental Assisting to complete competencies and necessary work experience.

### Limitations

The novel and innovative nature of the co-design framework means it remains a relatively untested methodology and lacks a robust tool for evaluating the efficacy of a protocol. Co-design requires elements of study design to remain intentionally imprecise and uncertain until the YAG starts meeting and the community co-design workshops take place. Elements of the project design may be altered as it is carried out to adapt to the revealed strengths and weaknesses and ensure the needs of individual communities are met. This uncertainty limits our ability to predict and mitigate potential complications which may arise.

The use of co-design methodology requires considerable preparation during the planning phase; raising the project’s initial cost and increasing the time before any benefits of the implementation phase can be felt. However, this also increases the likelihood of producing an effective oral health program that achieves its preventative health aims sooner. Aboriginal people vary considerably in culture and way of life around Australia. A successful oral health program developed with Aboriginal adolescents in one community may not be generalisable to other Aboriginal adolescents in other regions, limiting the wider utility of the study. However, the process of oral health co-design will be widely generalisable and, should the process prove successful, could be expanded for use as a model in other regions.

The flexible structure of the study increases its vulnerability to the influence of bias. Recall bias may be present in the self-reporting of the oral health habits of adolescents. Adolescents may overestimate or fail to remember their baseline oral health habits whilst completing the questionnaire. Furthermore, after forming a rapport with study investigators, adolescents may overestimate any improvements in oral health habits made. However, other studies have found this not to be the case [29]. Selection bias may affect the recruitment of Aboriginal adolescents. Aboriginal adolescents who are more outgoing and engaged with their local ACCHS would be more likely to be informed about and willing to participate in the study.

## 4. Conclusions

This study will provide a framework for engaging Aboriginal adolescents and their communities to achieve positive oral health outcomes. This study will provide health benefits to Aboriginal adolescent participants and have flow-on effects as positive oral health habits are shared between family members and peers. The success of this co-design preventative health protocol could lead to broader health benefits by influencing the direction of future government health initiatives and policy decisions.

## Figures and Tables

**Figure 1 ijerph-19-09104-f001:**
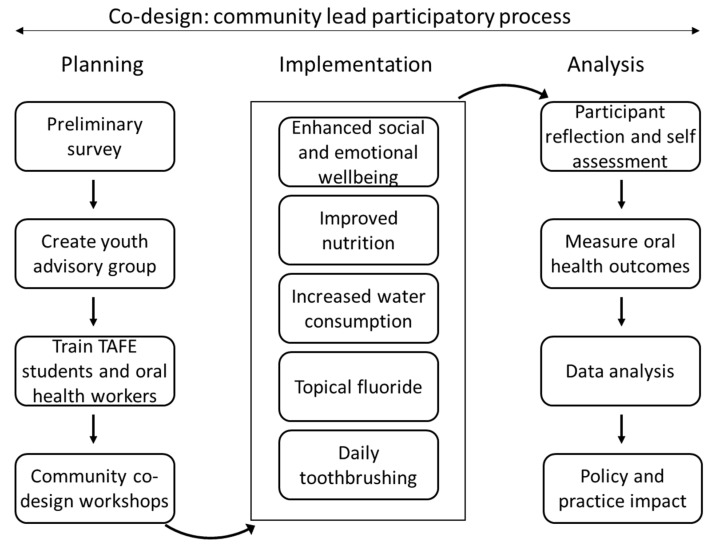
Phases of the Indigenous Adolescent Oral Health Partnership Study.

## Data Availability

Not applicable.

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
