# Peer review of "The Indigenous Adolescent Oral Health Partnership Study: A Co-Design Study Protocol"

_ijerph, 2022, doi:10.3390/ijerph19159104_

Round 1

Reviewer 1 Report

I must congratulate with the authors for their excellent work!!!

I have only one suggestion, even if this a protocol for a framework, I suggest to add a sample size calculation and some hypothesis on the outcome

There also some few minor misspelling 

Author Response

Thank you for your comments on this manuscript. Please see below for our responses.

I must congratulate with the authors for their excellent work!!!

Response: Thank you.

I have only one suggestion, even if this a protocol for a framework, I suggest to add a sample size calculation and some hypothesis on the outcome

Response: Thank you for this suggestion. Our sample size justification has been added (line 273) as well as the hypothesis (line 169).

There also some few minor misspelling 

Response: Thank you we have gone through the entire manuscript with editor in Microsoft Word and corrected any spelling errors.

Reviewer 2 Report

The Introduction provides an excellent overview of the health issues pertaining to a very important community that may be perceived as marginalised. The Introduction provides an impetus as to why the study needs to take place.

This article appears optimistic and promising with its positive forecasts about how the study will be conducted and what is to be expected. However, the article would be much worthier for publication if it had been actually implemented and evaluated with actual results and pertinent discussions.

If the study had been implemented in some form, if some results were actually obtained and analysed, and if learning points could be delivered, then this paper could surely be considered worthy for publication.  After the excellent Introduction, the study mainly consists on the future tense: lots of “..will.” Henceforth, at this stage, I must regrettably recommend a rejection.

Author Response

Reviewer 2:

The Introduction provides an excellent overview of the health issues pertaining to a very important community that may be perceived as marginalised. The Introduction provides an impetus as to why the study needs to take place.

This article appears optimistic and promising with its positive forecasts about how the study will be conducted and what is to be expected. However, the article would be much worthier for publication if it had been actually implemented and evaluated with actual results and pertinent discussions.

If the study had been implemented in some form, if some results were actually obtained and analysed, and if learning points could be delivered, then this paper could surely be considered worthy for publication.  After the excellent Introduction, the study mainly consists on the future tense: lots of “..will.” Henceforth, at this stage, I must regrettably recommend a rejection.

Response: Thank you for reviewing this article and providing your comments. The editors have proceeded with accepting this publication pending revisions. This study has not yet been implemented as it is a protocol paper. Therefore these suggestions cannot be included in the manuscript.

Reviewer 3 Report

Dear Authors,

I would like to appreciate the great efforts authors have made to produce such as masterpiece of work and i must say that it worths publishing. However, i have few suggestions.

1. Please mention the study timeline.

2. Please use the full form of the Acronym, when write for the first time such as NHMRC, ACCHS, NSW etc.

3. Did you take permission to use previously validated questionnaire, if yes, then mention..

4. What is certificate IV for Dental Assisting?

5. Do you have a plan to perform any pilot study?

6. How would you avoid the recall bias in the present study?

7. Please mention the Titanium software details such as the name of the Company, City, Country.

8. Source of funding for this project is missing? Please mention

9. What would be the gender of participants? In equal number?

10. Do you have a room for the modification of the study protocols if you face any unexpected situation?

11. How did you calculate the sample size? Do you think the 60 participants will represent the whole adolescent Aboriginal population?

12. How would to maintain the inter-examiner reliability? Do you have any plan for calibration? If yes, then How?

13. If any participant decide to withdraw his consent in the middle of the study then what would you do?

14. Please write the source of the the video mention on page 4 line 153?

15. Do you have any plan for auditing this project in future?

Author Response

Reviewer 3:

Thank you for reviewing this article and providing your comments. Our responses are listed below each point.

I would like to appreciate the great efforts authors have made to produce such as masterpiece of work and i must say that it worths publishing. However, i have few suggestions.

  1. Please mention the study timeline.

Response: Thank you, a timeline for this study has been added (line 166).

  1. Please use the full form of the Acronym, when write for the first time such as NHMRC, ACCHS, NSW etc.

Response: Thank you, we have gone through the manuscript to ensure that each time an acronym is used it is spelled out in full the first time.

  1. Did you take permission to use previously validated questionnaire, if yes, then mention..

Response: Yes, participants that completed the initial questionnaire signed a consent form. This has been added in the manuscript (158).

  1. What is certificate IV for Dental Assisting?

Response: A certificate IV in Dental Assisting is a vocational training qualification to be trained as and qualify as a dental assistant in Australia. A dental assistant is the same as a dental nurse that assists dental practitioners such as dentists and dental hygienists to complete dental treatment. This has been clarified in the manuscript in line 180.

  1. Do you have a plan to perform any pilot study?

Response: No.

  1. How would you avoid the recall bias in the present study?

Response: We plan to minimise recall bias by using an existing validated oral health questionnaire as a guide. Further, we will work with the YAG to ensure the questionnaire is culturally safe and relevant which will assist in minimising recall bias. These plans have been added in line 293.

  1. Please mention the Titanium software details such as the name of the Company, City, Country.

Response: This has been added in line 288.

  1. Source of funding for this project is missing? Please mention

Response: The source of funding for this project is not missing. We have written Not Applicable in Line 419 as we do not currently have a source of funding for this project, hence why we have published this protocol to assist with actively trying to secure a source of funding.

  1. What would be the gender of participants? In equal number?

Response: We are not specifying any gender or number of participants for each gender. This is because we have agreed in consultation with our Aboriginal co-investigators to set a sample size of 60 participants to recruit a diverse group of adolescents, including gender, age and location. Recruiting a diverse group of participants can be found in line 274.

  1. Do you have a room for the modification of the study protocols if you face any unexpected situation?

Response: Yes, most definitely. As this protocol will use co-design methodology, co-design allows for reflection and changes throughout the entire process to ensure that programs meet the needs and expectations of the community. This is has been clarified in the manuscript in line 306.

  1. How did you calculate the sample size? Do you think the 60 participants will represent the whole adolescent Aboriginal population?

Response: This number was agreed in consultation with our Aboriginal co-investigators to enable a diverse group of adolescents, including gender, age and location. This has been clarified in line 273.

  1. How would to maintain the inter-examiner reliability? Do you have any plan for calibration? If yes, then How?

In order to maintain inter-examiner reliability only registered dental practitioners will collect data on oral health status (added in line 286) and co-researchers who administer questionnaires will complete a skill set in community research. This qualification will cover basic research methodologies and data collection to enable co-researchers to collect data systematically.  This information has been added in line 183.

  1. If any participant decide to withdraw his consent in the middle of the study then what would you do?

Response: Participation in this study is completely voluntary. If a participant decides to withdraw in this study at any time, it will not affect their relationship with the investigators, their community or their local Aboriginal Community Controlled Health Service. This is already clearly written in the manuscript in line 282.

  1. Please write the source of the the video mention on page 4 line 153?

Response: The source of this video has been added in line 154.

  1. Do you have any plan for auditing this project in future?

Response: We will conduct a comprehensive evaluation of this project during the analysis phase which is described in section 2.8 of the manuscript.

Round 2

Reviewer 2 Report

Following on from my previous review, I have no further comments to make.